# Fatty Acid Profile and Antioxidative Properties of Peptides Isolated from Fermented Lamb Loin Treated with Fermented Milk

**DOI:** 10.3390/antiox9111094

**Published:** 2020-11-07

**Authors:** Małgorzata Karwowska, Anna D. Kononiuk, Dariusz M. Stasiak, Krzysztof Patkowski

**Affiliations:** 1Department of Meat Technology and Food Quality, University of Life Sciences in Lublin, ul. Skromna 8, 20-704 Lublin, Poland; malgorzata.karwowska@up.lublin.pl (M.K.); anna.kononiuk@up.lublin.pl (A.D.K.); 2Institute of Animal Breeding and Biodiversity Conservation, Faculty of Biology, Animal Sciences and Bioeconomy, University of Life Sciences in Lublin, 20-704 Lublin, Poland; krzysztof.patkowski@up.lublin.pl

**Keywords:** fermented lamb, fatty acid profile, CLA, peptide, antioxidant properties

## Abstract

This study evaluated the impact of fermented milk maceration on fermented lamb loin without nitrate to obtain peptides with high activity against oxidative changes (ABTS, DPPH, reducing power) as well as a favorable fatty acid profile, including CLA content. Additionally, an attempt was made to evaluate the influence of the lamb breed on the assessed properties. Raw loins (*m. Longissimus dorsi*) obtained from sheep of three polish breeds—Wrzosówka, Uhruska, and Świniarka—and fermented products were tested. The fermented loins obtained after 14 days of processing were characterized by pH and water activity values in the ranges, respectively, 4.76–5.12 and 0.902–0.915. The maceration of meat in a fermented milk has caused greater acidification of the meat during fermentation. Statistical analysis indicated that treatment was the factor with significant effect on peptide content; no effect of animal breed was found. The peptide content isolated from raw meat ranged from 2.90 to 4.31 mg g^−1^ of sample, while in fermented meat products it was significant higher (11.59–16.37 mg g^−1^ of product). The antioxidant properties of peptides were positively correlated with peptides content. The maceration in fermented milk resulted in a statistically significant increase of ABTS value in case of fermented lamb loin of Świniarka breed. The raw meat and fermented meat products form the Świniarka lamb breed were characterized by the highest content of the total CLA isomers. The main CLA isomer found was cis9-trans11 (rumenic acid), followed by cis9-cis11, trans9-trans11, and trans10-cis12. The rumenic acid content was higher than, respectively, 87% and 80–88% of total CLA isomers in case of raw meat and fermented lamb loins of three breeds.

## 1. Introduction

Fermented meat products have a far-reaching tradition and are preferentially produced and eaten in the Western world [1]. During their production, the main components of meat undergo multi-stage biochemical (proteolysis, lipolysis, oxidative changes) and physical (diffusion, water evaporation) changes, which shape the safety, quality and sensory features of the product, such as taste, color and aroma. Fermented meat products are appreciated by consumers not only for their unique taste and aroma, but also for their health benefits. Several studies have revealed that these products are the source of biologically active peptides [2,3,4]. The presence of these compounds is associated with the fermentation process, which results in proteolytic changes in proteins leading to the formation of peptides and amino acids. Our previous study [5], which determined the content of bioactive compounds in fermented sausages, including bioactive peptides, confirmed this relationship. Bioactive peptides are a specific protein fragments that could be generated through the action of endogenous enzymes during processing mainly fermentation [6]. Due to their positive impact on human health, many studies have recently been shown on the properties of peptides isolated from food, also in the context of meat and processed meat products [2,5]. The results of the research [7,8] showed antihypertensive, antioxidant, antibacterial and antiproliferative properties of peptides isolated from meat. Escudero et al. [9] conducted studies on the properties of a peptide extract isolated from Spanish dry-cured ham using the size-exclusion chromatography method. The antioxidant properties of some peptide fractions towards 1,1-diphenyl-2-picrylhydrazyl radical have been demonstrated. The peptides tested in this work also showed significant antihypertensive activity after oral administration to spontaneously hypertensive rats.

Fermented meat products are unique in so far as high-quality raw meat is used and synthetic additives are reduced or eliminated in the direction of the “clean label” trend. Such products are much preferred by modern consumers than those produced traditionally with synthetic additives, including nitrates [10]. Fermented ruminant whole meat products are usually made from beef; however, fermented lamb meat products can represent an interesting option in order to favor sheep sector. The more that lamb meat is characterized by exceptional nutritional values resulting from the fatty acid profile of intramuscular fat, including a high content of unsaturated acids and conjugated linoleic acid [11]. Previous studies reported that breed or genotype is an important factor influencing lamb meat quality [12,13]. The results obtained by Janiszewski et al. [14] also proved that the lamb breeds had an influence on the tested physical traits (e.g., pH, color, and tenderness) of the *longissimus lumborum* muscle of Polish native breeds including Uhruska, Wrzosówka, Świniarka that are the valuable local sheep breeds in Poland [15].

Additives are another important factor influencing the quality of fermented meat products. To obtain clean label products, natural raw materials of plant or animal origin are used, which show similar properties to synthetic additives [16]. Elimination of sodium nitrate, which shows antioxidant properties in meat products, creates the need to look for natural ingredients that have antioxidant properties. In this context, efforts to increase the content of bioactive compound in meat products, such as peptides with antioxidant properties, seem to be particularly important. As reported by Wang et al. [17], the role of bioactive peptides with antioxidant properties may be the protection of food against reactive oxygen species by capturing free radicals and chelating metal ions. Due to the presence of lactic acid bacteria and bioactive peptides, fermented milk can be an interesting addition in the production of raw meat products fermented with the use of native microflora [3]. Milk proteins are a source of bioactive peptides that are released during food processing [18] and show mineral-binding, opioid, ACE-inhibitory, immunomodulatory, cytotoxicity, anti-carcinogenic, antibacterial, and anti-thrombotic activities [19].

To the best of our knowledge, no studies have yet been published on the effect of fermented milk on the content and antioxidant properties of peptides isolated form fermented lamb loin. In this context, this study aims to assess the impact of fermented milk maceration on fermented lamb loin without nitrate in order to obtain peptides with high activity against oxidative changes (ABTS, DPPH, reducing power) as well as a favorable fatty acid profile, including CLA content. Additionally, an attempt was made to evaluate the influence of the lamb breed on the assessed properties.

## 2. Materials and Methods

### 2.1. Raw Materials

The raw materials for the production of fermented meats were loins (m. Longissimus dorsi), obtained from lambs of three polish breeds: Wrzosówka, Uhruska, and Świniarka. The animals were kept in farm conditions that met the requirements of organic farming. Uhruska lambs were slaughtered at the age of 5–6 months (the weight of the animals was 36 kg), while the lambs of the Wrzosówka and Świniarka breed were slaughtered at the age of 7–8 months and weigh 27–29 kg. Sheep for the experiment were from husbandry under the supervision of national veterinary inspection. The breeding met all the conditions required by European law. All animal rights under European law were controlled and respected at all stages from breeding, through transport to slaughter. The meat selected for experiment was purchased at butcher shop of the slaughterhouse. The meat was vacuum packed and delivered in cooling conditions to the laboratory was stored at 4 ± 1 °C for 48 h. After this time, the meat was prepared by cleaning the surface of the fascia, remaining tendons, giving the spindle shape a portion of about 1.0 kg. The meat samples prepared in this way were subjected to technological procedures, including salting (2.8%) with sea salt or curing salt (95.6% sea salt, 0.4% sodium nitrate (V), and maceration in a natural fermented milk for 48 h. Sea salt (non-iodinated and without anti-caking agents) (CuroDiMare, Italy) and glucose (Delecta, Poland) were purchased from local supermarkets (Lublin, Poland). Sodium nitrate (without anti-caking agents) was obtained from StanLab (Lublin, Poland). Fermented milk obtained from cow’s milk was bought fresh from certified diary product plant (R. Janowski, Ludwinów, Poland). In the next step, pieces of meat have been dried in ripening chamber at 18 °C for 24 h. The next day, the cement has been done, consisting of mustard seed, garlic, fresh red pepper paprika, black pepper, cumin, coriander and water. All spices have been placed in a blender and mixed until they form a cohesive dough. Then, pieces of meat have been covered by a paste of spices from all sides properly without leaving a gap to maintain the product. Batch variants were prepared by aging in fermentation chambers for 14 days under controlled humidity (75 ± 5%) and temperature (18 ± 1 °C) conditions. As a result, three types of treatment were obtained: P1—treatment with curing salt (95.6% sea salt, 0.4% sodium nitrate), P2—treatment with sea salt, and P3—treatment with sea salt and macerated in natural fermented milk.

### 2.2. Sampling and Experimental Design

Samples were taken from each variant at the end of processing. In addition, analyses were carried out for the raw material at the beginning of the experiment. The variants were replicated twice by producing two different batches on separate days. Eighteen fermented loins were produced in each batch. Both the raw material and the fermented product were tested for their physicochemical properties, peptides content and their antioxidant properties and fatty acids profile including CLA acids. All measurements were made in triplicate for each sample.

### 2.3. Physicochemical Properties (Water Activity, pH)

Water activity analyser (Novasina AG, Lachen, Switzerland), which gives temperature-controlled measurements, was used to measure water activity (a_w_). The device was calibrated with Novasina SAL-T humidity standards based on saturated salt solutions (33%, 75%, 84%, and 90% relative humidity).

The pH value was measured in a slurry obtained by homogenization 10 g of a minced sample with 50 mL of de-ionized water for 1 min using a homogenizer IKA T25 (IKA^®^-Werke GmbH & CO. KG, Staufen, Germany). A digital pH meter (CPC-501, Elmetron, Zabrze, Poland) equipped with pH electrode (ERH-111, Hydroment, Gliwice, Poland) and a temperature sensor was used.

### 2.4. Determination of Peptides Content and Their Antioxidative Properties

The peptide extraction was performed according to the method described by Zhu et al. [20]. The obtained supernatant was concentrated in an evaporator and dissolved in 0.01 M HCl and filtered through a 0.45 μm nylon membrane filter (AlfaChem, Toruń, Poland). The concentration of peptides was determined using o-phthaldialdehyde (OPA) spectrophotometric assay according to procedure by Nielsen et al. [21]. Leucine was used as a standard to quantify the peptides content. The peptide content was expressed as mg of peptides per 100 g of meat product.

The free radical scavenging activity of the peptide extract was determined using the ABTS (2-Azino-bis-3-ethylbenzothiazoline-6-sulfonic acid) method according to procedure described by Re et al. [22]. The scavenging activity of the peptides was expressed as Trolox equivalent mM per mg of peptides.

Antioxidative properties of peptides were also determined as DPPH radical scavenging activity according to the procedure described by Zhu et al. [20]. The ability of peptides to scavenge the DPPH free radicals was evaluated with reference to Trolox standard curve. The results were expressed as Trolox equivalent mM per mg of peptides.

Ferric-reducing antioxidant power (RP) of the peptides was determined according to Mora et al. [23]. The ability of peptides to reduce iron from the Fe^3+^ (ferric) oxidation state to the Fe^2+^ (ferrous) oxidation state was calculated in reference to the results obtained for ascorbic acid standard curve. The results were expressed as equivalent mg ascorbic acid per mg of peptides.

### 2.5. Fatty Acids Profile

To extract fat from the sample the mixture of chloroform and methanol at a ratio of 2:1 (*v*/*v*) was used as the extraction solvent according to the method of Folch et al. [24]. Gas chromatography analysis was performed using Varian 450-GC (Walnut Creek, CA, USA). Samples were separated using a capillary column (Select Biodiesel for FAME, Varian, Palo Alto, CA, USA) (30 m × 0.32 mm × 0.25 μm film thickness). Helium was used as the carrier gas at a flow rate of 1 mL per min. The initial temperature was set to 60 °C. After injection, the column temperature was programmed to rise 200 °C (maintained for 10 min), and subsequently increased to 240 °C at the rate of 3 °C min^−1^. The final temperature was held for 4 min. Reference standard methyl esters of fatty acids were used to identify the peaks.

### 2.6. Statistical Analysis

The experiment was replicated twice by producing two different batches on separate days. The Statistica v. 13.3 software was used to perform the statistical analysis of the results obtained in the experiment. In the manuscript data were expressed as the mean ± standard deviation. The experiment was conducted in two batches, the differences between the batches were not significant. The main effects (breeds and treatment) and interactions between means were calculated using factorial ANOVA. Post hoc comparisons were made using Tukey’s test. All differences were significant at *p* ≤ 0.05. To evaluate the strength of the relationship between variables Pearson’s correlation coefficient was calculated. Results were shown as a heat map of correlation.

## 3. Results and Discussion

### 3.1. pH and Water Activity Values

Table 1 shows the pH and water activity values of lamb raw meat and fermented products. The observations indicated that treatment was the only factor with significant effect on physicochemical characteristics; no effect of animal breed was found. Significance levels showed by experimental factors and their interactions for analyzed parameters of lamb raw meat and products are shown in Table 2. The obtained values of technological quality indicators (pH and a_w_ in the range of 5.51–5.59 and 0.972–0.974, respectively) were typical for good-quality fresh meat. As reported by Malva et al. [25], the variability of glycogen level before slaughter is one of the most important factors that affects the pH meat, which is typical for younger lambs susceptible on the stress. The pH values significantly decreased during the ripening process what is most likely related to the accumulation of the lactic acid produced in the fermentation. Villalobos-Delgado et al. [26] obtained much higher pH values for dry-cured lamb legs with different tumbling treatments after salting. Similarly, water activity significantly decreased after processing due to drying process. Statistical analysis showed the effect of treatments on pH values of fermented lamb loins after production. P3 sample (treatment with sea salt and fermented milk) was characterized by the lowest pH values compared to treatment with sea salt and sodium nitrate (P1) and only with sea salt (P2). The maceration of meat in a natural fermented milk has caused greater acidification of the meat during fermentation. Fermented milk is a source of lactic acid bacteria and some organic acids which can assist the loins fermentation process. Lactic acid is produced by fermentation of lactose, the concentration of some organic acids (lactic, propionic, acetic) increases [27].

### 3.2. Peptide Content and Its Antioxidants Properties

Table 3 shows the results of peptide content and antioxidant activity of peptides obtained from raw meat and fermented lamb loin. The results revealed that fermented meat products had several times higher content of peptide compared to raw meat. The peptide content isolated from raw meat ranged from 2.90 to 4.31 mg g^−1^ of sample while in fermented meat products it was in the range of 11.59 to 16.37 mg g^−1^ of product). These results are not surprising as peptides are the results of muscle protein degradation because of proteolytic activity of endogenous enzymes together with lactic acid bacteria. In fermented meat products, the protein degradation is influenced by different variables including product formulation and processing conditions, which also influence the obtained acidity of the products [28]. Kato et al. [28] point out that the presence of lactic acid bacteria induces a decrease of pH resulting in a greater activity of endogenous muscle proteases. This observation has been confirmed in this study. The correlation between the examined parameters, presented in the heat map (Figure 1), shows that the correlation between pH of fermented loins and the peptide content is negative. Thus, it was found that the peptide content increased with the decrease in the pH of the products. The statistical analysis also indicated that treatment was the factor with significant effect on peptide content, no effect of animal breed was found. In case of fermented loins produced from meat of Wrzosówka lamb breed, the highest content of peptides was found in the sample macerated with fermented milk (P3). It can be assumed that the use of fermented milk reduced the pH during fermentation of meat products and thus a higher activity of endogenous muscle proteases. Fermented milk itself is also a source of bioactive peptides [19,29]. As reported by El-Salam and El-Shibiny [30], peptides isolated from cow milk proteins show mineral-binding, opioid, ACE inhibitory, immunomodulatory, cytotoxicity, anti-carcinogenic, antibacterial, and anti-thrombotic activities. Whey proteins (lactalbumin, lactoperoxidase, lactoglobulin, immunoglobulins and lactoferrin) play a special role in the generation of bioactive peptides in fermented milk [3]. In addition, it was indicated that the sample of fermented lamb loin without the addition of sodium nitrate (P2) showed a higher content of bioactive peptides (14.41–15.80 mg g^−1^ of product) compared to the sample with nitrate (P1) (14.08–15.56 mg g^−1^ of product). The inverse relationship was found for Uhruska lamb meat products, the lowest peptide content was observed for the sample macerated with fermented milk. In the case of lamb meat products of the Świniarka breed, no statistically significant differences in the peptide content were found between the samples.

The mechanism of the antioxidant activity of bioactive peptides is not sufficiently elucidated; however, the commonly accepted method for testing antioxidant activity focuses on donating a hydrogen atom to a free radical as well as chelating metals to prevent or remove excess free radicals. In the present study, three radical scavenging capacity assays were applied to investigate antioxidant properties of peptides isolated from fermented lamb loins samples (ABTS and DPPH radical scavenging activity and ferric-reducing antioxidant power). Based on the results of statistical analysis presented in Table 2, the influence of experimental factors (breeds and treatment) and their interactions on the antioxidant properties of peptides was shown. Comparing the ABTS, DPPH and RP results of fermented lamb loins samples with and without nitrate, no significant differences were observed except for the DPPH values of the sample from lamb meat of the Wrzosówka breed. The DPPH antioxidant activity was higher in P1 with nitrate content (1.196–1.408 mg Trolox eqv./1 mg of peptide) sample compared to P2 sample without nitrate content (1.000–1.280 mg Trolox eqv./1 mg of peptide). Similar results were obtained in previous studies [31] that aimed at determining the effect of sodium nitrite reduction on the antioxidant properties of peptides isolated from cooked meat products. In a current study, maceration in fermented milk resulted in a statistically significant increase of ABTS value in case of fermented lamb loin of Świniarka breed. Similarly, fermented lamb loins of Świniarka and Uhruska breed, in which maceration of fermented milk was used, were characterized by significantly higher DPPH values (respectively, 1.499 and 1.492 mg Trolox eqv./1 mg of peptide) compared to P2 samples (respectively, 1.280 and 1.187 mg Trolox eqv./1 mg of peptide). The studies by Kęska and Stadnik [2] did not show the influence of the fermented milk product (acid whey) on the antioxidant properties of peptides isolated from dry-cured pork loin.

The correlation shown (Figure 1) in this study between the content of peptides and their antioxidant properties seems to be obvious. In the available literature, we can find examples showing this relationship. According to Karami and Akbari-Adergani [32] antioxidant activity of a peptide depends on the type of amino acids and the position that they occupy in the sequence. This relationship was also confirmed by the studies of Cumpy et al. [33]. They indicated that tri-peptides with tryptophan and tyrosine at their C-terminus showed strong antioxidant activity. Moreover, antioxidant activities were also dependent on different combinations of amino acids in tri-peptide chains. Other authors [34] in research on the antioxidant properties of peptides derived from tryptic hydrolysate of jumbo squid (Dosidicus gigas) skin gelatin demonstrated that the presence of hydrophobic amino acids in the peptide structure determines its antioxidant activity. In turn, the research of Rajapakse et al. [35] on the antioxidative properties of peptides from fermented mussel sauce indicated that aromatic amino acids improve the radical scavenging properties of the peptides because they have the ability to donate protons to electron deficient radicals. The authors also showed that antioxidative activity of peptides with histidine in the structure has attributed to the hydrogen-donating, lipid peroxyl radical trapping and/or the metal ion-chelating ability of the imidazole group.

### 3.3. Fatty Acid Profile and Conjugated Linoleic Acid Isomers Content

Table 4 presents the fatty acid composition (%) of raw meat and fermented lamb loins. SFA had the largest share in the fatty acid profile, followed by MUFA and then PUFA. Among SFA, C16:0 and C18:0 were the main fatty acids and their content in muscle fat differed slightly depending of the breed of lamb and ranged from 15.4% (for P3 fermented lamb loin of Świniarka breed) to 26.7% (for P1 and P3 fermented lamb loin of Wrzosówka breed (Figure 2. These findings were like those reported by Serra et al. [36] for lambs slaughtered at different weights.

In our study, linoleic acid was the main polyunsaturated fatty acid, ranging from 3.57% in fatty acid profile for raw loin of Świniarka breed to 6.74% in fatty acid profile for fermented loin of Świniarka breed. The content of total polyunsaturated fatty acids reached lower values than those reported by Kawecka et al. [15] for Wrzosówka lambs fed a linseed-supplemented diet. No significant differences in the PUFA content were observed for raw meat and fermented products made from it, with one exception. The P3 sample from the meat of the Świniarka lambs was characterized by a much higher content of the PUFAs compared to the rest treatments.

In terms of nutritional important ratios, the fermented lamb loins were observed to be beneficially lower than the maximum 4:1 ratio for n6/n3 in case of Świniarka breed, but higher for Wrzosówka and Uhruska breed. Additionally, the values of 0.13–0.32 for the ratio of PUFA/SFA obtained in our study was analogous to the value (0.28–0.37) for male suckling Massese lamb meat as presented by Serra et al. [36]. Similar results were also obtained by Baldi et al. [37] for *longissimus lumborum* muscle of lambs fed grain-based diets with moderate level of antioxidant, supra-nutritional level of antioxidant or lucerne hay-based diet. They got n−6/n−3 values in the range 2.47–3.30 and for PUFA/SFA ratio values from 0.16 to 0.24. It is less than the 0.4 minimum PUFA/SFA nutritional recommendations for ruminant meats [38].

Ruminants’ meat including lamb is an important source of conjugated linoleic acid [39] showing anti-cancer, anti-atherosclerotic and anti-diabetic properties and limiting the synthesis of adipose tissue [40]. The main CLA isomer found in raw meat and fermented lamb loins was cis9-trans11 (rumenic acid), followed by cis9-cis11, trans9-trans11, and trans10-cis12 (Figure 3). In case of raw meat, rumenic acid content was higher than 87% of total CLA isomers for the meat of each of the three breeds of lamb. The results are consistent with those obtained by Serra et al. [36] who observed that the content of this acid was greater than 87% of the total CLA isomers for *Longissimus dorsi*, *Triceps brachii* and *Semimembranosus muscles* for lambs slaughtered at 11–17 kg live weight. The present study indicated that the production process and fermentation reduced the content of this acid to such an extent that the fermented lamb loins were characterized by the rumenic acid content in the range of 80–88% of the total CLA isomers. As in the previous study [41], in which acid whey was used, no effect of fermented milk was found on the content of CLA. However, the highest content of the sum of CLA isomers was found for the raw meat and fermented meat products of the Świniarka lamb breed compared to those obtained from Wrzosówka and Uhruska breeds.

## 4. Conclusions

The results of this study demonstrated that fermented lamb loins, in which nitrate was eliminated during the production and fermented milk maceration was applied, are capable of retaining and delivering the nutritionally beneficial bioactive peptides and conjugated linoleic acid isomers. The study showed that the use of technological treatments affects the peptides content and its antioxidant properties. It seems that the meat of each of the studied breeds of lambs is valuable in the aspect of obtaining fermented products that are a rich source of peptides showing antioxidant properties. However, the most beneficial profile of fatty acids and the highest antioxidant activity of peptide extracts were obtained in the case of products made of the meat of the Świniarka breed.

## Figures and Tables

**Figure 1 antioxidants-09-01094-f001:**
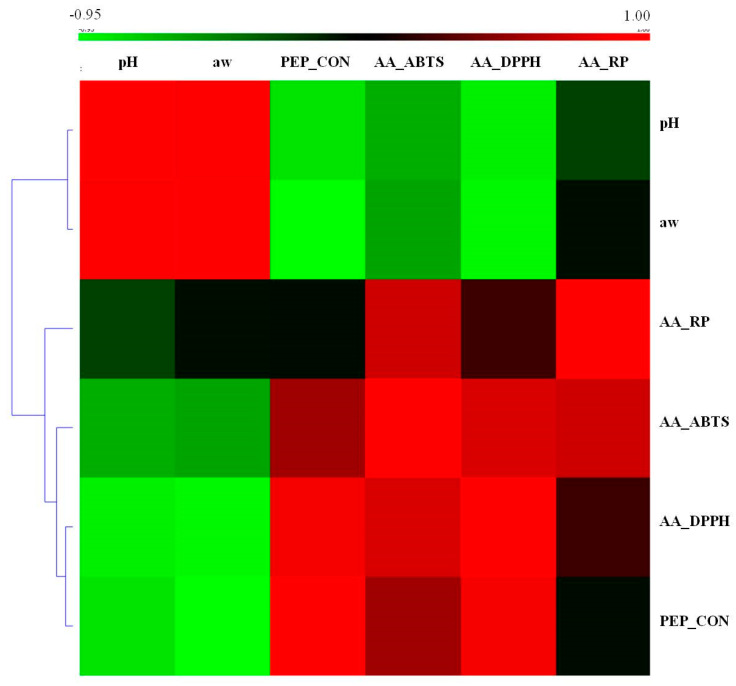
The correlations (shown as heat map) between research parameters.

**Figure 2 antioxidants-09-01094-f002:**
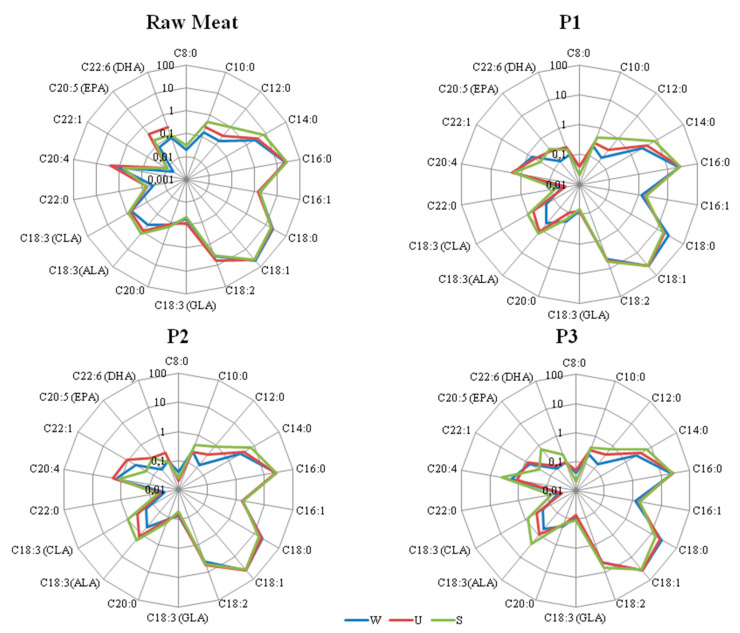
Fatty acids profile [%] of different treatment of lamb products. P1–P3 types of treatment: P1—treatment with sea salt and sodium nitrate, P2—treatment with sea salt, P3—treatment with sea salt and fermented milk. W, U, S—lamb breeds: W—Wrzosówka, U—Uhruska, S—Świniarka.

**Figure 3 antioxidants-09-01094-f003:**
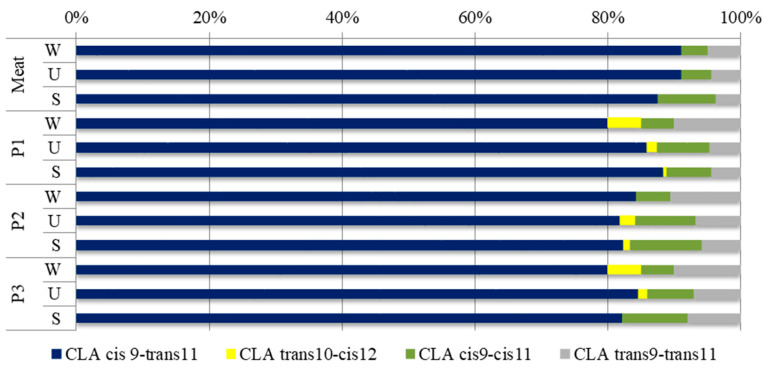
The contribution of individual CLA isomers to the total quantity of conjugated linoleic acid. W, U, S—lamb breeds: W—Wrzosówka, U—Uhruska, S—Świniarka; P1–P3 types of treatment: P1—treatment with sea salt and sodium nitrate, P2—treatment with sea salt, and P3—treatment with sea salt and fermented milk.

**Table 1 antioxidants-09-01094-t001:** pH and water activity of lamb raw meat and products (mean ± standard deviation).

Parameter	Lamb Breeds	Raw Meat	P1	P2	P3
a_w_	W	0.972 ^a^ ± 0.007	0.908 ^bcd^ ± 0.003	0.903 ^bc^ ± 0.003	0.913 ^bc^ ± 0.005
	U	0.974 ^a^ ± 0.001	0.910 ^bcd^ ± 0.002	0.902 ^c^ ± 0.001	0.915 ^b^ ± 0.001
	S	0.972 ^a^ ± 0.007	0.908 ^bcd^ ± 0.002	0.905 ^bc^ ± 0.002	0.907 ^bcd^ ± 0.001
pH	W	5.58 ^a^ ± 0.03	5.12 ^b^ ± 0.04	4.98 ^c^ ± 0.01	4.82 ^d^ ± 0.04
	U	5.59 ^a^ ± 0.09	5.12 ^b^ ± 0.03	5.02 ^bc^ ± 0.05	4.78 ^d^ ± 0.01
	S	5.51 ^a^ ± 0.02	5.10 ^b^ ± 0.02	5.05 ^bc^ ± 0.02	4.76 ^d^ ± 0.01

Means with different letters (^a–d^) differ significantly (*p* < 0.05). P1–P3 types of treatment: P1—treatment with sea salt and sodium nitrate, P2—treatment with sea salt, P3—treatment with sea salt and fermented milk, W, U, S—lamb breeds: W—Wrzosówka, U—Uhruska, S—Świniarka.

**Table 2 antioxidants-09-01094-t002:** Significance levels showed by experimental factors and their interactions for analyzed parameters of lamb raw meat and products.

Parameter	pH	a_w_	Peptide Content	Antioxidant Activity		
				ABTS	DPPH	RP
Breeds (B)	–	–	***	***	***	***
Treatment (T)	***	***	***	***	***	**
B × T	–	–	***	***	***	***

Fixed effects: B × T—interaction between breeds and treatment. *p*-value: *** (*p* < 0.001), ** (*p* < 0.01), * (*p* < 0.05); – (not significant).

**Table 3 antioxidants-09-01094-t003:** Peptide content and antioxidant activity of lamb raw meat and products.

Parameter	Lamb Breeds	Raw Meat	P1	P2	P3
peptide content mg g^−1^ of product	W	4.31 ^f^ ± 0.45	11.59 ^e^ ± 0.45	15.80 ^abc^ ± 0.88	16.37 ^a^ ± 0.73
	U	4.26 ^f^ ± 0.56	15.56 ^abc^ ± 0.45	16.06 ^ab^ ± 0.95	11.84 ^de^ ± 0.61
	S	2.90 ^g^ ± 0.03	14.08 ^bc^ ± 0.21	14.41 ^abc^ ± 1.22	13.88 ^cd^ ± 1.19
ABTS mg Trolox eqv. /1 mg of peptide	W	0.964 ^d^ ± 0.095	0.796 ^def^± 0.051	0.851 ^de^ ± 0.044	0.891 ^d^ ± 0.040
	U	0.614 ^f^ ± 0.067	1.610 ^bc^ ± 0.043	1.703 ^bc^ ± 0.080	1.801 ^b^ ± 0.098
	S	0.637 ^ef^ ± 0.085	1.645 ^bc^ ± 0.022	1.536 ^c^ ± 0.027	2.081 ^a^ ± 0.133
DPPH mg Trolox eqv. /1 mg of peptide	W	0.343 ^e^ ± 0.085	1.276 ^bc^ ± 0.090	1.000 ^d^ ± 0.008	1.164 ^cd^ ± 0.104
	U	0.117 ^f^ ± 0.050	1.196 ^c^ ± 0.042	1.187 ^cd^ ± 0.022	1.492 ^a^ ± 0.061
	S	0.064 ^f^ ± 0.007	1.408 ^ab^ ± 0.014	1.280 ^bc^ ± 0.077	1.499 ^a^ ± 0.104
RP mg ascorbic acid eqv./1 mg of peptide	W	0.024 ^de^ ± 0.002	0.020 ^e^ ± 0.001	0.020 ^e^ ± 0.000	0.020 ^e^ ± 0.001
	U	0.024 ^cde^ ± 0.001	0.028 ^bcd^ ± 0.003	0.030 ^bcd^ ± 0.001	0.033 ^ab^ ± 0.002
	S	0.031 ^abc^ ± 0.005	0.029 ^bcd^ ± 0.002	0.029 ^bcd^ ± 0.002	0.037 ^a^ ± 0.003

Means with different letters (^a–g^) differ significantly (*p* < 0.05). P1–P3 types of treatment: P1—treatment with sea salt and sodium nitrate, P2—treatment with sea salt, P3—treatment with sea salt and fermented milk, W, U, S—lamb breeds: W—Wrzosówka, U—Uhruska, S—Świniarka.

**Table 4 antioxidants-09-01094-t004:** Fatty acid composition (%) of raw meat and fermented lamb loin. P1–P3 types of treatment: P1—treatment with sea salt and sodium nitrate, P2—treatment with sea salt, and P3—treatment with sea salt and fermented milk.

	W				U				S			
	Raw Meat	P1	P2	P3	Raw Meat	P1	P2	P3	Raw Meat	P1	P2	P3
SFA	48.52	53.45	50.52	53.45	48.95	49.72	48.24	53.41	53.44	51.70	51.37	47.64
MUFA	45.22	39.35	43.02	39.35	40.34	40.62	42.08	39.59	39.97	38.03	39.25	37.38
PUFA	6.25	7.20	6.46	7.20	10.70	9.65	9.67	7.00	6.59	10.26	9.38	14.98
UFA/SFA	1.06	0.87	0.98	0.87	1.04	1.01	1.07	0.88	0.88	0.94	0.95	1.10
MUFA/SFA	0.93	0.74	0.85	0.76	0.82	0.82	0.87	0.75	0.75	0.74	0.77	0.79
PUFA/SFA	0.13	0.13	0.13	0.13	0.22	0.20	0.20	0.14	0.13	0.20	0.19	0.32
PUFA 6/3	8.82	8.68	8.50	8.66	5.55	4.29	4.32	5.53	2.76	3.68	2.97	3.16

SFA—saturated fatty acids; MUFA—monounsaturated fatty acids; PUFA—polyunsaturated fatty acids.

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
