# Peer review of "Fatty Acid Profile and Antioxidative Properties of Peptides Isolated from Fermented Lamb Loin Treated with Fermented Milk"

_antioxidants, 2020, doi:10.3390/antiox9111094_

Round 1

Reviewer 1 Report

The proposal of elimination of nitrates from the treatments and its substitution for natural ingredients, as the fermented milk, for the obtention of peptides with antioxidant properties and products with a favorable fatty acids profile, including CLA,  is interesting. However, the manuscript must be improved. I include some suggestions and comments. 

  • In the Introduction is not clear why the NaCl effect is adverse.
  • As a curiosity, did the authors think about the use of other type of milk, instead of the cow one or milk without lactose or proteins, for avoidind the problems of allergy of many people to this type of milk?
  • In the part of Statistical analysis, the authors say that the differences between the batches were not significant, where can it be observed?
  • Although, the different types of treatment (P1, P2 and P3) are known, I think that more explanation in the Methods section is necessary.
  • There is a confusion with the tables and figures: There are two Tables 2, Figure 2 is put after Figure 3 and there is no Figure 4, therefore, the discussion of results is difficult to follow. It is mandatory to correct it.
  • In Fig. 2, the term PC must be substituted by P3.
  • In Table 3, the results for Fatty acid composition of fermented lamb loin for W didn't change between P1 and P3, except for MUFA/SFA (0.74 instead of 0.76) and PUFA 6/3 (8.68 against 8.66). Is it possible?
  • The authors concluded that the effect of animal breed is not significant, but in the results in Table 2. Peptide content and antioxidant activity I can observe different values for peptide content, ABTS and DPPH with the lamb breeds in some of the cases.
  • Where the authors put acorbic and 15,4 and 26,7% is ascorbic, 15.4 and 26.7%, respectively.

Author Response

Dear Reviewer

We would like to thank the Reviewer for all valuable notes and comments. We have revised the manuscript according to recommendation. We have attached below a list of changes. Changes in the text are marked in red. We are convinced that Reviewer’s suggestions made the article much more valuable.

The proposal of elimination of nitrates from the treatments and its substitution for natural ingredients, as the fermented milk, for the obtention of peptides with antioxidant properties and products with a favorable fatty acids profile, including CLA,  is interesting. However, the manuscript must be improved. I include some suggestions and comments. 

In the Introduction is not clear why the NaCl effect is adverse.

The part of the introduction, which refers to the negative influence of salt in the context of the cited research, has been modified. Indeed, the quotation of the adverse effect of salt here is not related to the topic of the publication.

As a curiosity, did the authors think about the use of other type of milk, instead of the cow one or milk without lactose or proteins, for avoidind the problems of allergy of many people to this type of milk?

Thank you for this important observation. As the reviewer rightly noted, milk is an allergenic ingredient. The authors plan to use different milk than cow's milk in the production of fermented meat products, especially lamb (including sheep's milk). However, in these studies, it was decided to use cow’s milk because of its general availability also to industry.

In the part of Statistical analysis, the authors say that the differences between the batches were not significant, where can it be observed?

This analysis was performed as a preliminary one to exclude the influence of this factor. Therefore, after finding no significance, this factor was differentiated in the further analysis of the results. Only the results of the analysis of variance with factors (breeds and treatment) are presented in the manuscript.

Although, the different types of treatment (P1, P2 and P3) are known, I think that more explanation in the Methods section is necessary.

According to Reviewer’s suggestions additional information is provided at the end of section 2.1.

There is a confusion with the tables and figures: There are two Tables 2, Figure 2 is put after Figure 3 and there is no Figure 4, therefore, the discussion of results is difficult to follow. It is mandatory to correct it.

The authors apologize for this mistake. They made appropriate corrections in the text of the manuscript.

In Fig. 2, the term PC must be substituted by P3.

The authors apologize for the mistake. It has been corrected.

In Table 3, the results for Fatty acid composition of fermented lamb loin for W didn't change between P1 and P3, except for MUFA/SFA (0.74 instead of 0.76) and PUFA 6/3 (8.68 against 8.66). Is it possible?

I checked the results and they were not mistaken. The differences in the results in the second decimal place result from the inclusion of values with more digits after the decimal point in the calculations.

The authors concluded that the effect of animal breed is not significant, but in the results in Table 2. Peptide content and antioxidant activity I can observe different values for peptide content, ABTS and DPPH with the lamb breeds in some of the cases.

Table 2 presented significance levels showed by experimental factors and their interactions for analyzed parameters of lamb raw meat and products. The obtained data show that breeds had no statistically significant effect only in the case of pH and aw. There was a mistake with the peptide content, it was corrected.

Where the authors put acorbic and 15,4 and 26,7% is ascorbic, 15.4 and 26.7%, respectively.

The authors corrected the indicated typing error.

Reviewer 2 Report

This manuscript described the fatty acid profile and antioxidative properties of peptides isolated from fermented lamb loin treated with feremnted milk. All data presented in manuscript were oganized well. It clearly showd antioxidant properties of peptides derived from the fermented meat.

However it should revise the manuscript in the section of antioxidant activity measurement with units. Moreover the positve control experments with a certain known compound should be included in the Table 3.

Author Response

Dear Reviewer

We would like to thank Reviewer for all valuable notes and comments. We have revised the manuscript according to recommendation. We have attached below a list of changes. Changes in the text are marked in blue. We are convinced that Reviewer’s suggestions made the article much more valuable.

This manuscript described the fatty acid profile and antioxidative properties of peptides isolated from fermented lamb loin treated with feremnted milk. All data presented in manuscript were oganized well. It clearly showd antioxidant properties of peptides derived from the fermented meat.

However it should revise the manuscript in the section of antioxidant activity measurement with units. Moreover the positve control experments with a certain known compound should be included in the Table 3.

In the indicated part of the manuscript, descriptions were supplemented with values and units. Table 3 presented peptide content and antioxidant activity of lamb raw meat and products. Appropriate units have been applied.

Round 2

Reviewer 1 Report

The manuscript has been improved and it can be published.